# HIDING IMAGES IN DIFFUSION MODELS BY EDITING LEARNED SCORE FUNCTIONS

## ABSTRACT

Hiding data in deep neural networks (DNNs) has achieved remarkable successes, including both discriminative and generative models. Yet, the potential for hiding images in diffusion models remains underdeveloped. Existing approaches fall short in extracting fidelity, secrecy, and efficiency. In particular, the intensive computational demands of the hiding process, coupled with the slow extraction due to multiple denoising stages, make these methods impractical for resource-limited environments. To address these challenges, we propose hiding images at a specific denoising stage in diffusion models by modifying the learned score functions. We also introduce a parameter-efficient fine-tuning (PEFT) approach that combines parameter selection with a variant of low-rank adaptation (LoRA) to boost secrecy and hiding efficiency. Comprehensive experiments demonstrate the effectiveness of our proposed method.

## 1 INTRODUCTION

Data hiding involves concealing secret messages within various forms of cover media, such as bit streams (Cox et al., 2007), texts (Jassim, 2013), audios (Li & Yu, 2000), images (Baluja, 2019), videos (Swanson et al., 1997), and neural networks (Adi et al., 2018; Zhang et al., 2020). The goal is to minimize distortion to the cover media while ensuring the accurate retrieval of the hidden messages. Much like other areas in signal and image processing, data hiding has seen notable progress, especially with the integration of Deep Neural Networks (DNNs). Typically, DNN-based methods employ an autoencoder architecture, where the encoding network embeds the secret message into some cover media and the decoding network retrieves it. However, this scheme of *hiding data with DNNs* presents several limitations. First, the need to transmit the decoding network through a secure subliminal channel diminishes its practicality. Second, current DNN-based detection methods (Boroumand et al., 2018; You et al., 2020) have a good chance of detecting the presence of hidden data, undermining its secrecy. Last, hiding multiple secret messages for different recipients using the same encoding and decoding networks has proven challenging, thereby limiting its flexibility.

In stark contrast, the paradigm of *hiding data in DNNs* (Liu et al., 2020; Wang et al., 2021a;b; Chen et al., 2022; Fei et al., 2022; Zhang et al., 2024) has immediately improved secrecy due to the lack of detection methods taking DNN weights as input. Additionally, it has eliminated the need for sharing a decoding network between the sender and receiver with enhanced practicality. Initially, the focus was primarily on hiding data in discriminative models (Liu et al., 2020; Wang et al., 2021a;b), but there has been a shift towards using generative models for this purpose as their utility has become more recognized. Certain existing approaches (Chen et al., 2022; Fei et al., 2022; Zhang et al., 2024) have effectively concealed images in GANs (Goodfellow et al., 2014). However, these techniques are insufficient and, in some cases, infeasible for diffusion models (Song & Ermon, 2019; 2020; Song et al., 2021; Ho et al., 2020; Dhariwal & Nichol, 2021; Karras et al., 2022), primarily due to two reasons: 1) The neural network in diffusion models learns the score function (Song et al., 2020) of the data distribution, which differs from the neural network generator in GANs that learns to generate data samples directly, 2) The generative process in diffusion models entails multi-step denoising utilizing a shared neural network, leading to a distinct training scheme that is more intricate and time-consuming compared to training GANs.

To address these challenges, recent approaches (Zhao et al., 2023; Peng et al., 2023; Xiong et al., 2023; Chou et al., 2024; Fernandez et al., 2023; Feng et al., 2024) have been proposed for hiding data in diffusion models. However, these methods have their limitations. Some (Zhao et al., 2023; Peng et al., 2023) require computationally intensive re-training of diffusion models, making them unfeasible in resource-constrained environments. Others (Xiong et al., 2023; Chou et al., 2024; Fernandez et al., 2023; Feng et al., 2024) use fine-tuning to reduce computational costs but still suffer from limited hiding capacity and efficiency. Additionally, they are exclusively applicable to latent diffusion models (Rombach et al., 2022), due to the necessity of modifying text input for text-to-image synthesis (Chou et al., 2024), latent representations produced by latent encoder(Feng et al., 2024), or latent decoder(Xiong et al., 2023; Fernandez et al., 2023).

This paper introduces a new approach for hiding images in diffusion models by modifying the learned score function, distinct from previous methods, focusing on a secret time step in the generative process (refer to Fig. 1), which offers increased flexibility, security, and efficiency. We introduce a secrecy loss that regulates performance degradation in the original generation task, employing a distillation framework with a pre-trained diffusion model as a reference model, as shown in Fig. 2b. Furthermore, we construct a hybrid parameter-efficient fine-tuning (PEFT) method by combining selective PEFT techniques (Han et al., 2024) with parameterized PEFT techniques (Kalajdzievski, 2023; Hayou et al., 2024), largely reducing the number of modified parameters and accelerating the hiding process. Experiments demonstrate the proposed method yields: 1) high fidelity, extracting the secret images with minimal distortion, 2) high secrecy, as the modified diffusion model behaves normally, 3) high efficiency, significantly reducing the time cost for hiding images in diffusion models, compared to existing methods, and 4) high flexibility, enabling the hiding multiple images for different receivers.

## 2 RELATED WORK

In this section, we provide a review of the pertinent research fields related to this paper, encompassing neural network steganography, diffusion models, and parameter-efficient fine-tuning.

**Neural Network Steganography (NNS).** Hiding data in neural networks can be applied in NNS (Song et al., 2017; Liu et al., 2020; Wang et al., 2021a;b; Chen et al., 2022), which involves concealing a secret message within a neural network for covert communication, striking a balance between extraction accuracy and secrecy. Various existing NNS methods employ diverse strategies to embed confidential information in the neural network. These strategies include replacing the least significant bits (LSB) of model parameters (Song et al., 2017; Liu et al., 2020), replacing selected redundant parameters (Liu et al., 2020; Wang et al., 2021a), mapping the values of model parameters to secret message (Song et al., 2017; Liu et al., 2020; Wang et al., 2021b), mapping the signs of model parameters to secret bits string (Song et al., 2017; Liu et al., 2020), memorizing arbitrarily labeled synthetic data whose labels encode secret information (Song et al., 2017), and concealing secret image in neural network-based probabilistic models (Chen et al., 2022). However, NNS with diffusion models has been relatively underexplored. This paper investigates the viability of diffusion model-based NNS by introducing an image-hiding method for diffusion models.

**Diffusion Models.** Diffusion models employ a manageable forward corruption process that can be reversed to generate data. The reverse process is learned by diffusion models, eliminating the need for carefully designed network architectures. UNet (Ronneberger et al., 2015) and Transformer (Vaswani et al., 2017) are commonly employed architectures for diffusion models. Diffusion models can be categorized into pixel-space diffusion models (Ho et al., 2020; Song et al., 2021; Karras et al., 2022) and latent-space diffusion models (Rombach et al., 2022; Peebles & Xie, 2023; Podell et al., 2024), depending on whether the diffusion process and its reverse apply to the pixels or the latent representations of the image. This paper specifically concentrates on concealing images in pixel-space diffusion models, particularly the Denoising Diffusion Probabilistic Model (DDPM) with UNet architecture, as an illustrative example.

**Parameter-Efficient Fine-Tuning (PEFT).** Parameter-efficient fine-tuning (PEFT) methods can be divided into reparameterized and selective fine-tuning (Han et al., 2024). Reparameterized fine-tuning introduces additional trainable parameters into the frozen pre-trained backbone. Low-rank

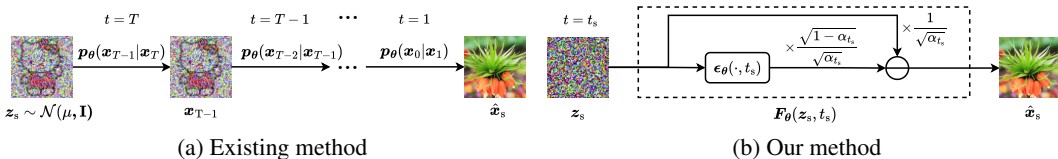

(a) Existing method              (b) Our method

Figure 1: Comparison of existing method and our method. (a) Existing methods introduce image patterns as secret keys at each denoising step to trigger a guided denoising process that reconstructs the secret image $x_s$. (b) Our method uses the fixed Gaussian noise $z_s$ and a secret time step $t_s$ as secret keys, which are utilized to reconstruct the secret image with $F_{\theta}(z_s, t_s)$.

adaptation (LoRA) and its variants (Hu et al., 2022; Nam et al., 2022; Edalati et al., 2022; Zhang et al., 2023a; Lialin et al., 2023; Kalajdzievski, 2023; Liu et al., 2024; Hayou et al., 2024; Meng et al., 2024; Wang & Liang, 2024; Wang et al., 2024) are representative reparameterized methods. Selective methods, on the other hand, carefully select a subset of parameters using various strategies (Zaken et al., 2021; He et al., 2023; Zhang et al., 2023b) for fine-tuning. This paper introduces a hybrid PEFT method for hiding images in diffusion models, combining the benefits of reparameterized and selective PEFT techniques.

## 3 METHOD

In this section, we first provide a preliminary discussion on diffusion models. Next, we describe the scenario for our method. Finally, we detail the proposed method for hiding images in diffusion models.

### 3.1 PRELIMINARY

Diffusion models are commonly used generative models for data generation, especially in image generation. The training and inference of diffusion models involve the diffusion process and the reverse process. In this subsection, we discuss these procedures using the DDPM (Ho et al., 2020) as an example.

**Diffusion Process.** The diffusion process gradually introduces Gaussian noise to a clean image $x_0$ sampled from the data distribution $\mathcal{P}_{\text{data}}$. This noise is added from time step $t = 1$ to $t = T$, with the variance of the noise determined by a predefined schedule $\beta_1, \ldots, \beta_T$. The diffusion process can be described as a Markov chain:

$$q\left(x_{1:T} \mid x_0\right) := \prod_{t=1}^{T} q\left(x_t \mid x_{t-1}\right), \tag{1}$$

$$q\left(x_t \mid x_{t-1}\right) := \mathcal{N}\left(x_t; \sqrt{1 - \beta_t} x_{t-1}, \beta_t \mathbf{I}\right). \tag{2}$$

At the final time step $t = T$, the clean image $x_0$ is diffused to Gaussian noise $x_T \sim \mathcal{N}(\mathbf{0}, \mathbf{I})$.

**Reverse Process.** In contrast to the diffusion process, the reverse diffusion process is a denoising process. It can be described by a Markov chain with learned Gaussian transitions parameterized by $\theta$:

$$p_{\theta}\left(x_{0:T}\right) := p\left(x_T\right) \prod_{t=1}^{T} p_{\theta}\left(x_{t-1} \mid x_t\right). \tag{3}$$

The training objective of $\theta$ is to minimize the variational bound on the negative log-likelihood, which is equivalent to minimizing the expression derived in (Ho et al., 2020):

$$\mathbb{E}_{t, x_t, \epsilon}\left[\left\|\epsilon - \epsilon_{\theta}\left(x_t, t\right)\right\|_2^2\right], \tag{4}$$

where $\epsilon \sim \mathcal{N}(\mathbf{0}, \mathbf{I})$, $\boldsymbol{x}_t = \sqrt{\bar{\alpha}_t}\boldsymbol{x}_0 + \sqrt{1 - \bar{\alpha}_t}\epsilon$, $\epsilon_{\boldsymbol{\theta}}$ is a neural network with parameter $\boldsymbol{\theta}$ for predicting $\epsilon$ from $\boldsymbol{x}_t$, $\bar{\alpha}_t = \prod_{i=1}^{t}(1 - \beta_i)$, and time step $t \sim \text{Uniform}(\{1, ..., T\})$. During inference, the estimation of clean image $\boldsymbol{x}_0$ from $\boldsymbol{x}_t$ can be derived as:

$$\boldsymbol{x}_0 = \frac{1}{\sqrt{\bar{\alpha}_t}}\left(\boldsymbol{x}_t - \sqrt{1 - \bar{\alpha}_t}\epsilon_{\boldsymbol{\theta}}\left(\boldsymbol{x}_t, t\right)\right). \tag{5}$$

This estimation process of $\boldsymbol{x}_0$ will be leveraged for hiding/extracting images in/from the diffusion models in the proposed method.

## 3.2 Image Hiding Scenario

Three entities are involved in the image-hiding scenario:

- The sender, who hides images within a pre-trained diffusion model and transmits the stego diffusion model.

- The receiver, who possesses the secret key to extract the hidden images from the received stego diffusion model.

- The inspector, who inspects the stego diffusion model before it reaches the receiver, using commonly used metrics to determine if the diffusion model meets expectations. If the stego diffusion model fails to meet the expectation, it will be considered suspicious and deleted.

In the given scenario, an image-hiding method should ensure: 1) The extracted secret image should closely resemble the ground truth secret image. 2) The generation performance of stego diffusion model should be similar to that of the original diffusion model.

## 3.3 Proposed Method

Previous studies (Zhao et al., 2023; Peng et al., 2023) modify the diffusion model to enable a guided reverse process, where the secret image is generated step by step using a predefined secret key (*e.g.*, a secret pattern blended with Gaussian noise), refer to Fig. 1a and Fig. 2a. However, such approaches result in inferior secret image reconstruction and notably reduce the original generation performance of diffusion models, particularly when the secret image is a natural image rather than a simple icon. Moreover, constructing a guided reverse process with a large number of steps results in slow extraction and hiding processes, resembling the standard inference and training processes of diffusion models.

Motivated by recent studies (Liu et al., 2023; Song et al., 2023; Sauer et al., 2023; Yin et al., 2023) that aim to accelerate the inference of diffusion models by distilling a one-step diffusion model from a pre-trained diffusion model, we propose hiding/extracting images in/from one secret time step of the denoising process. Our proposed hiding scheme allows the secret image to be concealed in an arbitrarily chosen secret time step, which serves as the component of the secret key along with the predefined fixed noise for secret image extraction, as shown in Fig 1b and Fig 2b.

### 3.3.1 Hiding Pipline

We denote the ground truth secret image as $\boldsymbol{x}_{\text{s}}$, the secret key as $\boldsymbol{z}_{\text{s}}$, and the secret time step as $t_{\text{s}}$. The one-step secret image reconstruction function is defined as:

$$\boldsymbol{F}_{\boldsymbol{\theta}}(\boldsymbol{z}_{\text{s}}, t_{\text{s}}) = \frac{1}{\sqrt{\bar{\alpha}_{t_{\text{s}}}}}\left(\boldsymbol{z}_{\text{s}} - \sqrt{1 - \bar{\alpha}_{t_{\text{s}}}}\epsilon_{\boldsymbol{\theta}}\left(\boldsymbol{z}_{\text{s}}, t_{\text{s}}\right)\right), \tag{6}$$

where $\epsilon_{\boldsymbol{\theta}}\left(\cdot, t\right)$ is the modified learned score function. The expression of $\boldsymbol{F}_{\boldsymbol{\theta}}(\boldsymbol{z}_{\text{s}}, t_{\text{s}})$ is adapted from the formula in Equation 5, which is originally used to estimate clean image $\boldsymbol{x}_0$ in the step $t$ of the reverse process.

Our objective is to modify the parameter $\boldsymbol{\theta}$ in the neural network $\epsilon_{\boldsymbol{\theta}}\left(\cdot, t\right)$ in order to ensure that the output of $\boldsymbol{F}_{\boldsymbol{\theta}}(\boldsymbol{z}_{\text{s}}, t_{\text{s}})$ is a reconstruction of $\boldsymbol{x}$s with minimal distortion. At the same time, we want to preserve the functionality of $\epsilon_{\boldsymbol{\theta}}\left(\cdot, t\right)$ in the standard denoising process of diffusion models. To achieve this objective, $\boldsymbol{\theta}$ can be optimized using a loss function with fidelity and secrecy loss terms.

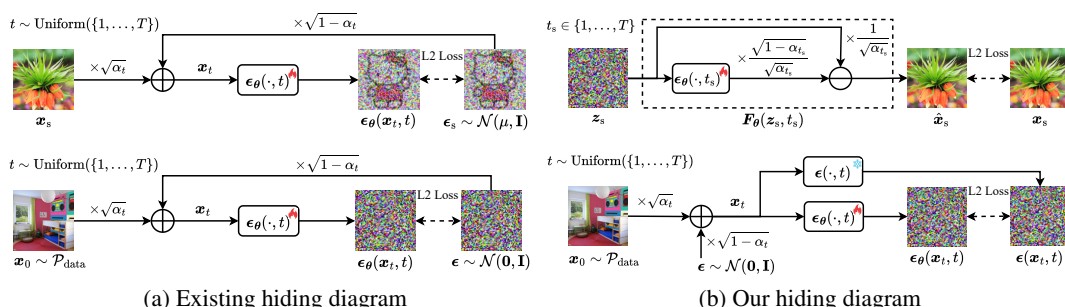

(a) Existing hiding diagram          (b) Our hiding diagram

Figure 2: Hiding diagrams of existing method and our method. (a) Existing methods train a guided denoising process (refer to the upper pipeline) resembling the standard training of diffusion models (refer to the lower pipeline). In the guided denoising process, all intermediate noise contains pre-defined secret patterns. Additionally, the standard training process of diffusion models is utilized in parallel as regularization. (b) Our method modifies the $\epsilon_{\boldsymbol{\theta}}(\cdot, t_s)$ to reconstruct the secret image with $\boldsymbol{F_\theta}(\boldsymbol{z}_s, t_s)$ only in one step (refer to the upper pipeline), with a regularization (refer to the lower pipeline) that forces the $\epsilon_{\boldsymbol{\theta}}(\cdot, t)$ to maintain functionality as the pre-trained score function $\epsilon(\cdot, t)$.

**Fidelity Loss.** The objective of the fidelity loss is to ensure that the secret image reconstruction $\hat{\boldsymbol{x}}_s = \boldsymbol{F_\theta}(\boldsymbol{z}_s, t_s)$ matches the ground truth secret image $\boldsymbol{x}_s$. Using the squared error as the distortion function, the fidelity loss is defined as:

$$\ell_a(\boldsymbol{x}_s, \boldsymbol{z}_s, t_s) = \|\boldsymbol{F_\theta}(\boldsymbol{z}_s, t_s) - \boldsymbol{x}_s\|_2^2. \tag{7}$$

With the fidelity loss, we can optimize the score function $\epsilon_{\boldsymbol{\theta}}(\cdot, t)$ to hide the secret image $\boldsymbol{x}_s$ with the secret key $\boldsymbol{z}_s$ and $t_s$, as illustrated in Fig 2b. However, the requirement of secrecy, which involves preserving the functionality of $\epsilon_{\boldsymbol{\theta}}(\cdot, t)$ in the standard denoising process, has not been fulfilled. Consequently, a regularization term is introduced to ensure secrecy.

**Secrecy Loss.** The objective of the secrecy loss is to maintain the functionality of $\epsilon_{\boldsymbol{\theta}}(\cdot, t)$ in the standard denoising process. Instead of relying on the standard training process of diffusion models as regularization (see Fig 2a), we choose to enforce the modified score function $\epsilon_{\boldsymbol{\theta}}(\cdot, t)$ to match the output of the pre-trained score function $\epsilon(\cdot, t)$ (see Fig 2b), given the same noisy image inputs at time step $t$. In practice, $\epsilon(\cdot, t)$ is a clone of the score function from the pre-trained diffusion model with fixed parameters. The secrecy loss is then defined as:

$$\ell_b(\boldsymbol{x}_t, t) = \mathbb{E}_{t, \boldsymbol{x}_t} \left[ \|\epsilon_{\boldsymbol{\theta}}(\boldsymbol{x}_t, t) - \epsilon(\boldsymbol{x}_t, t)\|_2^2 \right], \tag{8}$$

where $t \sim \mathrm{Uniform}(\{1, ..., T\})$ represents the uniformly sampled time step at each iteration, $\boldsymbol{x}_t \sim \mathcal{N}(\boldsymbol{x}_t; \sqrt{\bar{\alpha}_t}\boldsymbol{x}_0, (1 - \bar{\alpha}_t)\boldsymbol{I})$ is the noisy image in step $t$ of the diffusion process.

Finally, the total loss function $\ell$ is the sum of fidelity loss $\ell_a$ and secrecy loss $\ell_b$:

$$\ell = \ell_a(\boldsymbol{x}_s, \boldsymbol{z}_s, t_s) + \lambda \ell_b(\boldsymbol{x}_t, t), \tag{9}$$

where $\lambda$ is the trade-off parameter.

**Hiding Multiple Images.** Hiding multiple images for different recipients poses challenges, not only due to the increased amount of information to conceal, but also because each recipient should only be able to extract a specific concealed image, and should not be able to extract other concealed images (Wang et al., 2021b). Nevertheless, the proposed method can be extended to handle this challenging task. Specifically, given a set of secret images $\mathcal{X}_s = \left\{\boldsymbol{x}_s^{(1)}, \boldsymbol{x}_s^{(2)}, \ldots, \boldsymbol{x}_s^{(C)}\right\}$ where $\boldsymbol{x}_s^{(i)}$ denotes the $i$-th secret image, $C$ is the number of secret images to be hidden, and each $\boldsymbol{x}_s^{(i)}$ is paired with a secret key $\boldsymbol{z}_s^{(i)}$ from a set of secret keys $\mathcal{Z}_s = \left\{\boldsymbol{z}_s^{(1)}, \boldsymbol{z}_s^{(2)}, \ldots, \boldsymbol{z}_s^{(C)}\right\}$ shared to $C$ different receivers, we modify the loss function $\ell$ in Equation 9 to:

$$\ell_{\mathrm{multiple}} = \frac{1}{C} \sum_{i=1}^{C} \ell_a(\boldsymbol{x}_s^{(i)}, \boldsymbol{z}_s^{(i)}, t_s) + \lambda \ell_b(\boldsymbol{x}_t, t). \tag{10}$$

By optimizing the pre-trained diffusion model using $\ell_{\mathrm{multiple}}$, the $i$-th recipient is able to extract the secret image $\boldsymbol{x}_{\mathrm{s}}^{(i)}$ using the assigned secret key $\boldsymbol{z}_{\mathrm{s}}^{(i)}$. Without additional secret keys, the $i$-th recipient is unable to extract other secret images that are meant to be shared with other receivers.

### 3.3.2 PEFT METHOD

To enhance the efficiency of the hiding process and ensure better preservation of secrecy by modifying fewer model parameters, we propose a hybrid PEFT algorithm that combines the strengths of selective and reparameterized PEFT techniques. The proposed PEFT algorithm consists of three steps: computing parameter-level sensitivity, selecting the sensitive layers, and applying reparameterized PEFT to the sensitive layers.

**Parameter Sensitivity.** The importance of parameters in a pre-trained neural network for a specific task can be indicated by the parameter sensitivity $\boldsymbol{s}$, which is defined as the squared gradient of the loss function $\ell$ with respect to its parameter (He et al., 2023):

$$s_i = g_i^2, \tag{11}$$

where $i$ is the parameter index, $s_i$ is the $i$-th element of $\boldsymbol{s}$, $g_i$ is the $i$-th element of gradients $\boldsymbol{g} = \partial\ell/\partial\boldsymbol{\theta}$. We focus on parameter sensitivity because the highest sensitivity indicates the direction of the fastest convergence of the loss function, thereby aiding efficient gradient descent during fine-tuning. To obtain accurate parameter sensitivity, $\boldsymbol{s}$ is accumulated over $M$ iterations before fine-tuning, to create the accumulated sensitivity $\boldsymbol{S}$:

$$\boldsymbol{S} = \sum_{m=1}^{M} \boldsymbol{s}^{(m)}, \tag{12}$$

where $\boldsymbol{s}^{(m)}$ represents the $\boldsymbol{s}$ in the $m$-th iteration. We then select the top-$\tau$ largest elements from $\boldsymbol{S}$, where the parameter budget $\tau = \gamma N$, $N$ is the number of parameters in $\boldsymbol{\theta}$, and $\gamma \in [0,1]$ is a hyperparameter indicating the sparsity of the sensitive parameters. Finally, a set of sensitive parameters, denoted as $\boldsymbol{\theta}_{\mathrm{s}}$, is obtained using the indices of the top-$\tau$ largest elements from $\boldsymbol{S}$.

For PEFT, it is feasible to exclusively fine-tune the sensitive parameters $\boldsymbol{\theta}_{\mathrm{s}}$, as done in (He et al., 2023), by masking the gradients of non-sensitive parameters during updates using a gradient mask $\boldsymbol{M}$:

$$M_i = \begin{cases} 1 & \theta_i \in \boldsymbol{\theta}_{\mathrm{s}} \\ 0 & \theta_i \notin \boldsymbol{\theta}_{\mathrm{s}} \end{cases}, \tag{13}$$

where $M_i$ denotes the $i$-th element of $\boldsymbol{M}$, and $\theta_i$ represents the $i$-th element of $\boldsymbol{\theta}$. However, such a PEFT method does not eliminate the need to calculate the gradients of all parameters in $\boldsymbol{\theta}$, resulting in computational costs identical to those of full fine-tuning. To improve the fine-tuning efficiency, LoRA can be applied to selected layers of the neural network. By updating only the LoRA parameters, the number of learnable parameters is reduced, resulting in computational savings.

**Sensitive Layers Selection.** To determine the most appropriate layers to fine-tune, we select layers based on parameter sensitivity, referred to as sensitive layers. Specifically, given a set of sensitive parameters $\boldsymbol{\theta}_{\mathrm{s}}$, we analyze the distribution of sensitive parameters across each layer of the pre-trained neural network. We define the sensitive layers as the top-$n$ layers that contain the highest number of sensitive parameters, where $n = \delta K$, $K$ is the total number of layers in the pre-trained neural network, and $\delta \in [0,1]$ is a hyperparameter indicating the sparsity of sensitive layers. Once the sensitive layers are identified, we can proceed to apply LoRA to these selected layers.

**LoRA-based PEFT.** While numerous variants of LoRA techniques exist, offering improved efficiency over the basic approach, our method utilizes two variants of LoRA, namely rsLoRA (Kalajdzievski, 2023) and LoRA+ (Hayou et al., 2024), to achieve faster convergence speed. It is important to note that the sensitive layers in the UNet architecture of the diffusion models can encompass both linear layers and convolutional layers. Since standard LoRA-based techniques are restricted to linear layers, we employ the LoCon method from the LyCORIS library (Yeh et al., 2023), which extends the standard LoRA for convolutional layers.

Figure 3: Error maps of extracted secret image.

Table 1: Extraction fidelity comparison for $32 \times 32$ and $256 \times 256$ secret images. To measure fidelity, PSNR, SSIM, LPIPS, and DISTS are calculated between the extracted and ground truth secret images. "↑": larger is better, and vice versa. The top two methods are highlighted in boldface.

| Method | $32 \times 32$ | | | | $256 \times 256$ | | | |
|---|---|---|---|---|---|---|---|---|
| | PSNR↑ | SSIM↑ | LPIPS↓ | DISTS↓ | PSNR↑ | SSIM↑ | LPIPS↓ | DISTS↓ |
| Baluja17 | 25.40 | 0.89 | 0.116 | 0.051 | 26.46 | 0.90 | 0.200 | 0.089 |
| HiDDeN | 25.24 | 0.88 | 0.252 | 0.075 | 27.13 | 0.91 | 0.233 | 0.100 |
| Weng19 | 26.66 | 0.93 | 0.059 | 0.035 | 33.85 | 0.95 | 0.089 | 0.047 |
| HiNet | 30.39 | 0.94 | 0.033 | 0.026 | 35.31 | 0.96 | 0.087 | 0.041 |
| PRIS | 29.83 | 0.94 | 0.041 | 0.027 | **37.42** | **0.97** | **0.050** | **0.029** |
| Chen22 | **47.72** | **0.99** | **0.001** | **0.002** | 36.44 | 0.96 | 0.073 | 0.035 |
| BadDiffusion | 22.08 | 0.86 | 0.129 | 0.060 | 17.68 | 0.81 | 0.386 | 0.137 |
| TrojDiff | 46.54 | **0.99** | **0.001** | 0.004 | 24.74 | 0.94 | 0.057 | 0.076 |
| WDP | 36.49 | **0.99** | 0.003 | 0.008 | 17.97 | 0.83 | 0.245 | 0.144 |
| Ours | **52.90** | **0.99** | **0.001** | **0.001** | **39.33** | **0.97** | **0.043** | **0.018** |

## 4 EXPERIMENTS

### 4.1 EXPERIMENTAL SETUP

**Model and Dataset.**  We conduct experiments using DDPM, although our approach can be easily applied to other categories of diffusion models such as EDM (Karras et al., 2022) and consistency models (Song et al., 2023). The architecture and hyperparameters of DDPM follow the specification of Ho et al. (2020). The default secret time step $t_s$ for hiding/extracting the secret image is 500. For better demonstration, we not only hide low-resolution ($32 \times 32$) images in the DDPM pre-trained on CIFAR10 (Krizhevsky & Hinton, 2009) but also hide high-resolution images ($256 \times 256$) in the DDPM pre-trained on LSUN bedroom (Yu et al., 2015).

**Secret Image and Secret Key.**  Prior studies (Zhao et al., 2023; Peng et al., 2023) commonly employ secret images with simple content, such as QR code images, icon-like images, or images from MNIST/Fashion-MNIST datasets. However, we contend that simple secret images may not adequately demonstrate the effectiveness of the hiding approaches. Therefore, to address this limitation, we constructed a secret image dataset by aggregating images from several widely used natural image datasets, including COCO (Lin et al., 2014), DIV2K (Agustsson & Timofte, 2017), LSUN church (Yu et al., 2015), and Places (Zhou et al., 2018). The secret key is predefined to guide the modified score function $\epsilon_\theta$ in generating the secret image. In our experiments, the secret key is the fixed Gaussian noise randomly sampled from $\mathcal{N}(\mathbf{0}, \mathbf{I})$, which can be reproduced by specifying the seed of the pseudo-random generator for generating Gaussian noise.

### 4.2 EVALUATION METRICS

**Fidelity.**  Fidelity refers to the level of distortion (lower is better) or similarity (higher is better) between the extracted and ground truth secret images. To evaluate fidelity, we employ image quality metrics, including peak signal-to-noise ratio (PSNR), structural similarity index (SSIM) (Wang et al., 2004), learned perceptual image patch similarity (LPIPS) (Zhang et al., 2018), and deep image structure and texture similarity (DISTS) (Ding et al., 2020).

Table 2: Secrecy and hiding efficiency, when hiding $32\times32$ secret images. For secrecy measurement, FID is the population-level metric, while PSNR, SSIM, LPIPS, and DISTS are the sample-level metrics. Time is with respect to GPU hours, which measures hiding efficiency.

| Method | FID↓ | PSNR↑ | SSIM↑ | LPIPS↓ | DISTS↓ | Time↓ |
|---|---|---|---|---|---|---|
| Pretrained | 4.79 | N/A | N/A | N/A | N/A | N/A |
| BadDiffusion | 6.88 | 23.78 | 0.80 | 0.222 | 0.082 | 4.87 |
| TrojDiff | **4.64** | **28.72** | **0.91** | **0.114** | **0.049** | 12.72 |
| WDP | 5.09 | 22.50 | 0.84 | 0.228 | 0.083 | **2.35** |
| Ours | **4.77** | **31.06** | **0.94** | **0.077** | **0.037** | **0.04** |

Table 3: Secrecy and hiding efficiency, when hiding $256 \times 256$ secret images.

| Method | FID↓ | PSNR↑ | SSIM↑ | LPIPS↓ | DISTS↓ | Time↓ |
|---|---|---|---|---|---|---|
| Pretrained | 7.46 | N/A | N/A | N/A | N/A | N/A |
| BadDiffusion | 15.75 | 16.40 | 0.60 | 0.452 | 0.224 | 31.92 |
| TrojDiff | **14.36** | **18.73** | **0.70** | **0.407** | **0.169** | 83.68 |
| WDP | 15.07 | 18.59 | 0.67 | 0.445 | 0.200 | **11.36** |
| Ours | **8.39** | **23.31** | **0.83** | **0.235** | **0.112** | **0.18** |

**Secrecy.** For assessing the image generation performance of generative models, the commonly used metric is Fréchet inception distance (FID) (Heusel et al., 2017), measuring the quality of generated images at the population level. Previous studies calculate FID on the pre-trained diffusion model and the modified diffusion model, respectively, with the discrepancy indicating the level of secrecy. However, such a population-level image quality metric may not reflect the nuances in individual images. Therefore, we measure the secrecy at the individual sample level by computing the individual image distortion (indicated by PSNR, SSIM, LPIPS, and DISTS) between images generated from the pre-trained and edited diffusion models using the same initial noise.

**Hiding Efficiency.** We record the time cost (in terms of GPU hours) for the hiding process as an indicator of hiding efficiency. All methods in our experiment are implemented on the same device equipped with AMD EPYC 7F52 16-Core CPU and NVIDIA GeForce RTX3090 GPU.

### 4.3 EXPERIMENTAL RESULTS

**Fidelity.** Table 1 presents a quantitative comparison of our method with existing image steganography (Baluja, 2019; Zhu et al., 2018; Weng et al., 2019; Jing et al., 2021; Yang et al., 2024) and NNS (Chen et al., 2022) methods. Additionally, we include the results of applying existing backdoor attack methods (Chou et al., 2023; Chen et al., 2023) and a diffusion model watermarking method (Peng et al., 2023) for diffusion model-based NNS. Our method achieves the highest fidelity for both $32 \times 32$ and $256 \times 256$ resolution images. A qualitative comparison of the fidelity, in the form of an error map between the extracted and ground truth secret images with $256 \times 256$ resolution, is presented in Fig.3.

**Secrecy.** Table 2 and Table 3 provide a quantitative comparison of secrecy. Our method achieves the most similar FID to that of the pre-trained diffusion model, while also exhibiting the best PSNR, SSIM, LPIPS, and DISTS for individual samples. These results indicate that our method provides the best secrecy when hiding images in diffusion models. Fig. 4 visually demonstrates the qualitative comparison of sample-level distortion for the generated LSUN-bedroom images ($256 \times 256$). The secrecy evaluation of image steganography methods is based on the fidelity of the stego image, which is not directly comparable to the secrecy (fidelity of stego neural network models) of NNS methods. However, we still include the results in Appendix. The secrecy results for the NNS method are also included in the Appendix for clarity, as their stego neural network model is not diffusion model.

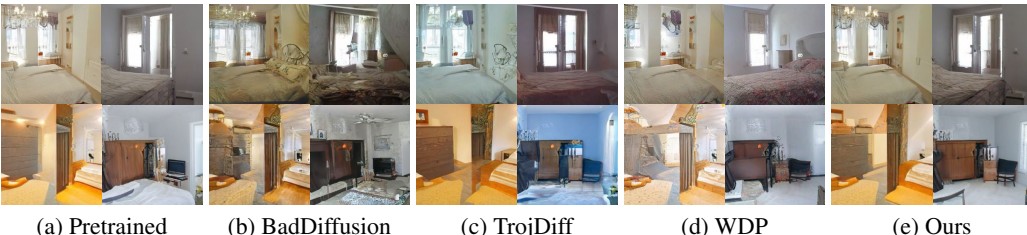

| (a) Pretrained | (b) BadDiffusion | (c) TrojDiff | (d) WDP | (e) Ours |

Figure 4: Qualitative comparison of generated samples.

Table 4: Fidelity and secrecy comparison, when hiding multiple $32 \times 32$ secret images.

| # of Images | Fidelity (Extracted Secret Image) | | | | Secrecy (Generated Image) | | | |
|---|---|---|---|---|---|---|---|---|
| | PSNR↑ | SSIM↑ | LPIPS↓ | DISTS↓ | PSNR↑ | SSIM↑ | LPIPS↓ | DISTS↓ |
| 1 | 52.90 | 0.99 | 0.001 | 0.001 | 31.06 | 0.94 | 0.077 | 0.037 |
| 4 | 49.38 | 0.99 | 0.001 | 0.001 | 30.93 | 0.95 | 0.064 | 0.033 |
| 8 | 43.11 | 0.99 | 0.001 | 0.002 | 30.59 | 0.94 | 0.076 | 0.038 |

Table 5: Fidelity and secrecy comparison, when hiding multiple $256 \times 256$ secret images.

| # of Images | Fidelity (Extracted Secret Image) | | | | Secrecy (Generated Image) | | | |
|---|---|---|---|---|---|---|---|---|
| | PSNR↑ | SSIM↑ | LPIPS↓ | DISTS↓ | PSNR↑ | SSIM↑ | LPIPS↓ | DISTS↓ |
| 1 | 39.33 | 0.97 | 0.043 | 0.018 | 23.31 | 0.83 | 0.235 | 0.112 |
| 4 | 38.31 | 0.96 | 0.058 | 0.029 | 17.78 | 0.74 | 0.394 | 0.165 |
| 8 | 33.55 | 0.91 | 0.161 | 0.066 | 14.08 | 0.62 | 0.510 | 0.187 |

**Hiding Efficiency.** Table 2 and Table 3 demonstrate that our method exhibits the lowest time cost (in terms of GPU hours), indicating superior hiding efficiency. Other compared methods require re-training or full fine-tuning for a large number of iterations, resulting in high time costs. Among them, WDP (Peng et al., 2023) has the lowest time cost, although it is still significantly higher compared to our method.

**Hiding Multiple Images.** Tables 4 and 5 show the fidelity and secrecy of our hiding method when multiple images are concealed. In general, as the number of secret images increases, the fidelity decreases. However, the overall image-hiding performance of our method remains acceptable. It can be observed that hiding low-resolution images is easier compared to high-resolution images, given the same number of secret images.

## 5 ABLATION EXPERIMENTS

**Different Secret Time Steps.** The proposed hiding scheme enables the hiding of a secret image in an arbitrarily selected secret time step $t_{\rm s}$, which acts as a component of the secret key. In this experiment, we investigate the impact of the selected secret time step $t_{\rm s}$ on the hiding performance. As depicted in Fig. 5, different choices of $t_{\rm s}$ result in similarly good hiding performance in terms of fidelity and secrecy (measured by PSNR and DISTS). More results can be found in Appendix. Notably, the highest fidelity is achieved when $t_{\rm s}$ falls between 700 and 900, and a better trade-off between fidelity and secrecy is observed.

**Comparison with Full Fine-tuning.** To demonstrate the effectiveness of our PEFT method, we compare it with full fine-tuning. Table 6 presents the results, showing that replacing our PEFT method with full fine-tuning leads to a noticeable decline in secrecy without enhancing fidelity. This highlights the advantage of our PEFT method, which improves secrecy without sacrificing fidelity due to its modification of fewer parameters for image hiding.

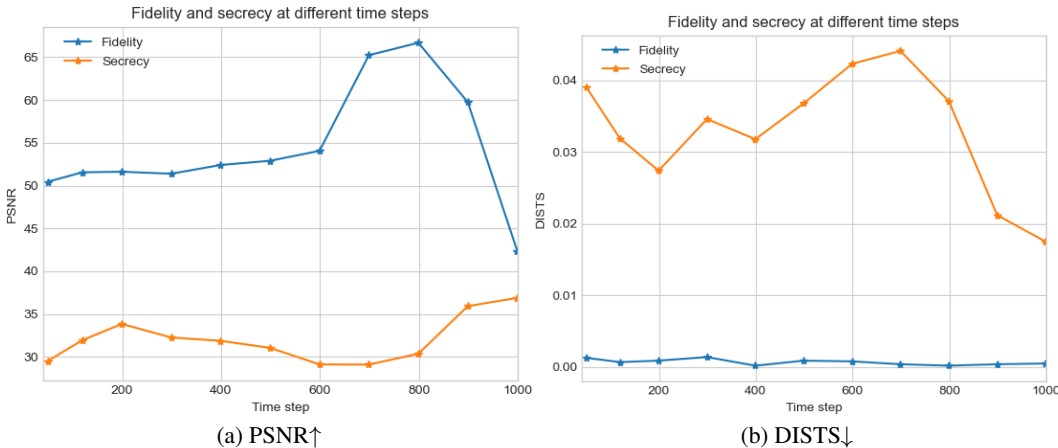

(a) PSNR↑                    (b) DISTS↓

Figure 5: Ablation of different selected time steps. The fidelity is indicated by blue lines and the secrecy is indicated by orange lines. More results can be found in the Appendix.

Table 6: Fidelity and secrecy comparison. FFT means replacing our PEFT with full fine-tuning.

| Method | Fidelity (Extracted Secret Image) | | | | Secrecy (Generated Image) | | | |
|---|---|---|---|---|---|---|---|---|
| | PSNR↑ | SSIM↑ | LPIPS↓ | DISTS↓ | PSNR↑ | SSIM↑ | LPIPS↓ | DISTS↓ |
| FFT ($32 \times 32$) | 51.30 | 0.99 | 0.001 | 0.001 | 22.53 | 0.83 | 0.200 | 0.075 |
| Ours ($32 \times 32$) | 52.90 | 0.99 | 0.001 | 0.001 | 31.06 | 0.94 | 0.077 | 0.037 |
| FFT ($256 \times 256$) | 38.36 | 0.97 | 0.057 | 0.022 | 17.87 | 0.65 | 0.474 | 0.197 |
| Ours ($256 \times 256$) | 39.33 | 0.97 | 0.043 | 0.018 | 23.31 | 0.83 | 0.235 | 0.112 |

Table 7: Fidelity and secrecy comparison, when hiding image in other types of diffusion models.

| Type | Fidelity (Extracted Secret Image) | | | | Secrecy (Generated Image) | | | |
|---|---|---|---|---|---|---|---|---|
| | PSNR↑ | SSIM↑ | LPIPS↓ | DISTS↓ | PSNR↑ | SSIM↑ | LPIPS↓ | DISTS↓ |
| DDPM | 52.90 | 0.99 | 0.001 | 0.001 | 31.06 | 0.94 | 0.077 | 0.037 |
| EDM | 51.30 | 0.99 | 0.002 | 0.001 | 28.80 | 0.94 | 0.090 | 0.056 |
| CM | 35.93 | 0.98 | 0.012 | 0.008 | 25.57 | 0.91 | 0.112 | 0.073 |

**Other Types of Diffusion Models.** While we have demonstrated the proposed method using DDPM as an example in this paper, it is worth noting that the proposed method can be applied to other types of diffusion models. In this ablation experiment, we extend the application of our method to hide images in EDM (Karras et al., 2022) and consistency model (Song et al., 2023). The pre-trained models utilized in this experiment include an EDM trained on ImageNet $64 \times 64$, and a consistency model distilled from such EDM. As shown in Table 7, satisfactory fidelity and secrecy are achieved when employing the proposed method for hiding images in other types of diffusion models.

## 6 CONCLUSION

In conclusion, this research addresses the limitations of current methods for hiding images in diffusion models by introducing a new approach that hides images at a secret time step in the denoising process, leveraging a hybrid PEFT method that combines the advantages of selective and reparameterized PEFT techniques to improve hiding efficiency. Extensive analyses and experiments demonstrate the superiority of the proposed approach in terms of fidelity, secrecy, and hiding efficiency. Future work could explore potential enhancements and applications of our approach in more advanced multi-modal diffusion models.

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
