# A APPENDIX

## A.1 SECRECY OF THE COMPARED IMAGE STEGANOGRAPHY AND NNS METHODS

Table 1: Secrecy of the compared image steganography methods. PSNR, SSIM, LPIPS, and DISTS are calculated between the ground truth cover images and the stego images.

| Method | $32 \times 32$ | | | | $256 \times 256$ | | | |
|---|---|---|---|---|---|---|---|---|
| | PSNR↑ | SSIM↑ | LPIPS↓ | DISTS↓ | PSNR↑ | SSIM↑ | LPIPS↓ | DISTS↓ |
| Baluja17 | 27.05 | 0.90 | 0.085 | 0.039 | 33.15 | 0.95 | 0.065 | 0.037 |
| HiDDeN | 28.72 | 0.95 | 0.116 | 0.043 | 30.44 | 0.96 | 0.147 | 0.057 |
| Weng19 | 35.87 | 0.97 | 0.030 | 0.021 | 36.00 | 0.96 | 0.041 | 0.031 |
| HiNet | 31.09 | 0.95 | 0.013 | 0.012 | 40.83 | 0.97 | 0.019 | 0.007 |
| PRIS | 32.34 | 0.96 | 0.007 | 0.008 | 41.32 | 0.98 | 0.005 | 0.006 |

Table 2: Secrecy of the compared NNS method. Single image Fréchet inception distance (SIFID) and diversity score (DS) are population-level metrics that measure the quality and diversity of a set of generated images, while PSNR, SSIM, LPIPS, and DISTS are sample-level metrics that measure the discrepancy between each two samples generated from the pre-trained and stego models respectively.

| Method | $32 \times 32$ | | | | | | $256 \times 256$ | | | | | |
|---|---|---|---|---|---|---|---|---|---|---|---|---|
| | SIFID↓ | DS↑ | PSNR↑ | SSIM↑ | LPIPS↓ | DISTS↓ | SIFID↓ | DS↑ | PSNR↑ | SSIM↑ | LPIPS↓ | DISTS↓ |
| Pretrained | 0.032 | 0.027 | N/A | N/A | N/A | N/A | 0.019 | 0.436 | N/A | N/A | N/A | N/A |
| Chen22 | 0.041 | 0.034 | 32.09 | 0.93 | 0.033 | 0.016 | 0.016 | 0.468 | 17.85 | 0.66 | 0.294 | 0.087 |

## A.2 MORE RESULTS OF TIME STEP ABLATION EXPERIMENT

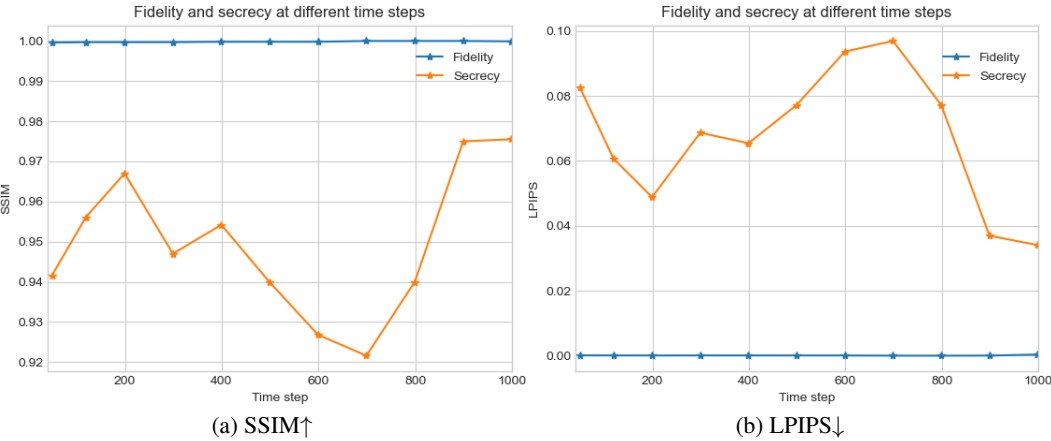

(a) SSIM↑      (b) LPIPS↓

Figure 1: Ablation of different secret time steps. The fidelity is indicated by blue lines and the secrecy is indicated by orange lines.