# OpenReview forum: "Hiding Images in Diffusion Models by Editing Learned Score Functions"
_ICLR.cc/2025/Conference — ICLR 2025 Conference Withdrawn Submission_

### Official Review · Reviewer_ByKt · 2024-10-29

**Soundness:** 2
**Presentation:** 2
**Contribution:** 2
**Rating:** 5
**Confidence:** 4

**Summary:**

This paper proposes a Neural Network Steganography method based on diffusion models. By finetuning the diffusion model to match the output given the secret key and secret timestep, this paper realizes steganography with high fidelity, high secrecy, high efficiency and high flexibility.

**Strengths:**

1. The idea is simple and the paper is easy to comprehend.
2. The paper introduces a method to select sensitive layers in PEFT, which may be effective in various tasks.

**Weaknesses:**

1. The biggest concern for me is the practical utility for Neural Network Steganography (which hide data in a neural network). What is the advantage to use a neural network as the container, rather than some other more lightweight container such as a single image?
2. Though a PEFT method is proposed to tackle the computational overhead and a secrecy loss is proposed to preserve the generative capalicity for benign samples, finetuning the neural network still seems not a good solution, as it will manipulate the parameters, making the approximated distribution inaccuracy. Why not fix the neural network and finetune the secret key $z_s$? Moreover, this paper aims to embed a specific image into the Unet of diffusion models. However, methods like ddim inversion or textual inversion can also embed a specific image back into the latent noise with almost perfect reconstruction accuracy. What is the advantage to finetune the parameters?
3. Though the method claim that multi-image hiding is supported, the experiments are conducted in a single image manner. More details about the experiments should be included: for example, the image amount of the hiding data, the amount of finetuning layers.
4. Small Typos:
* In line 214 $xs\rightarrow x_s$

**Questions:**

1. What is the FID for pretrained model in Table 2 and Table 3? Is it the divergence between generated image and the training dataset?
2. Ablations with smaller timestep interval should be conducted (Figure 5) Current ablation is under an interval of 100. This is beneficial to choose a better secret timestep. Besides, according to Figure 5, why choose 500 as the secret timestep?
3. The fidelity and secrety is satisfactory in the secret timestep. What is the performance of adjacent timesteps? Do they show similar performance? To be specific, choosing 500 as the secret timestep, the reconstruction of $x_{500\rightarrow 0}$ is fairy good. How about the reconstruction results of $x_{499\rightarrow 0}$ and $x_{501\rightarrow 0}$? Do they share similar reconstruction accuracy?

---

### Official Review · Reviewer_5gUL · 2024-11-01

**Soundness:** 3
**Presentation:** 3
**Contribution:** 3
**Rating:** 6
**Confidence:** 3

**Summary:**

Existing methods for data hiding in DNN are limited by fidelity, secrecy and efficiency, making them impractical for constrained environments.To address these issues, the authors propose a method that embeds images at a specific denoising stage in diffusion models by modifying learned score functions. They also introduce a parameter-efficient fine-tuning (PEFT) strategy that uses LoRA to enhance both secrecy and efficiency.

**Strengths:**

- The experimental results are quite impressive, significantly surpassing the current SOTA in terms of fidelity, secrecy, and efficiency.
- The presentation of the paper is excellent, being very clear and straightforward.

**Weaknesses:**

- The novelty is limited, as the paper's contributions are built upon existing algorithms. For instance, efficiency is achieved using PEFT, and the only truly innovative aspects are the proposed secrecy loss and the method for selecting sensitive layers for PEFT.
- One aspect I find unclear is why your PEFT method outperforms FFT in Table 6.
- The ablation study lacks an analysis of the impact of the \(\lambda\) parameter.

**Questions:**

- Why your PEFT method outperforms FFT in Table 6.

---

### Official Review · Reviewer_D4TF · 2024-11-03

**Soundness:** 2
**Presentation:** 2
**Contribution:** 2
**Rating:** 5
**Confidence:** 4

**Summary:**

The authors modify the learned score functions to hide images in diffusion models and introduce a parameter-efficient fine-tuning (PEFT) method to enhance security and efficiency. The authors demonstrate the effectiveness of the proposed method through extensive experiments.

**Strengths:**

1. The authors evaluated the method on several large-scale datasets, including COCO, DIV2K, and Places, providing a comprehensive testing platform to prove the method’s effectiveness.
2. The authors used a parameter-efficient fine-tuning (PEFT) method, significantly improving security and efficiency.
3. The paper is logically clear and easy to understand.

**Weaknesses:**

1. A simple application of the diffusion model is not novel enough.
2. The work lacks the contribution summary in the manuscript.
3. The designed hybrid PEFT method seems the recombination of existing algorithms.

**Questions:**

1. How to prove the uniqueness of the secret key? Authors should evaluate this property.
2. The effect of hyperparameters should be analyzed and added.
3. What is the difference between the designed hiding pipeline and the prior Formula 5? Does it bring any challenges when direct application?
4. The hybrid PEFT algorithm is the extension of prior methods. Thus, authors need to conduct a detailed ablation study to prove its effectiveness.
5. The comparison methods are not recently published SOTA methods (before 2023). Thus, it can’t prove its superior performance.

Overall, I think this paper can’t achieve the bar of a top-tier conference.

---

### Official Review · Reviewer_ES8H · 2024-11-04

**Soundness:** 2
**Presentation:** 2
**Contribution:** 1
**Rating:** 3
**Confidence:** 4

**Summary:**

Embedding data within deep neural networks (DNNs) has seen significant success, spanning both discriminative and generative models; however, image embedding within diffusion models remains underexplored. Current methods struggle with limitations in fidelity, secrecy, and efficiency, particularly due to the high computational cost of embedding and the slow extraction process required by multiple denoising stages, rendering these techniques impractical in resource-constrained settings. To overcome these challenges, this work proposes embedding images at a specific denoising stage within diffusion models by adjusting the learned score functions. Additionally, a parameter-efficient fine-tuning (PEFT) strategy, incorporating selective parameter tuning with a low-rank adaptation (LoRA) variant, is introduced to enhance both secrecy and efficiency. Experimental results substantiate the effectiveness of the proposed method.

**Strengths:**

1. This paper introduces a novel steganography method specifically designed for diffusion models.

**Weaknesses:**

1. The novelty of the proposed method is limited, as techniques like fine-tuning the U-Net of diffusion models to embed specific responses or remove certain content have already been extensively explored in related work (e.g., [1]), even though this reference is not cited.

2. Embedding information through the weights of diffusion models poses practical challenges; these weights typically range between 5GB and 10GB, which results in a considerably higher transmission load compared to image-domain steganography methods.

3. Approaches such as [2] that apply steganography directly within the parameter space of models should also be included in the comparisons to offer a more comprehensive evaluation.

4. Comparisons with image-based steganography methods may not be entirely fair, as these methods prioritize maintaining similarity between the cover and steganographic images, which could impact their performance metrics in steganographic capacity or secrecy.

5. Further evaluations on additional open-source diffusion models are recommended to strengthen the assessment and generalizability of the proposed approach.





[1] @InProceedings{Gandikota_2023_ICCV,
    author    = {Gandikota, Rohit and Materzynska, Joanna and Fiotto-Kaufman, Jaden and Bau, David},
    title     = {Erasing Concepts from Diffusion Models},
    booktitle = {Proceedings of the IEEE/CVF International Conference on Computer Vision (ICCV)},
    month     = {October},
    year      = {2023},
    pages     = {2426-2436}
}

[2] @inproceedings{cui2024steganographic,
  title={Steganographic Passport: An Owner and User Verifiable Credential for Deep Model IP Protection Without Retraining},
  author={Cui, Qi and Meng, Ruohan and Xu, Chaohui and Chang, Chip-Hong},
  booktitle={Proceedings of the IEEE/CVF Conference on Computer Vision and Pattern Recognition},
  pages={12302--12311},
  year={2024}
}

**Questions:**

See weakness above.

---

### Note · Authors · 2024-11-13

I have read and agree with the venue's withdrawal policy on behalf of myself and my co-authors.